# The DmeRF System Is Involved in Maintaining Cobalt Homeostasis in *Vibrio parahaemolyticus*

**DOI:** 10.3390/ijms24010414

**Published:** 2022-12-27

**Authors:** Yuxuan Zhao, Mengyao Kong, Jiaxue Yang, Xiaoxian Zhao, Yiran Shi, Yimeng Zhai, Jun Qiu, Chengkun Zheng

**Affiliations:** 1Jiangsu Key Laboratory of Zoonosis, Yangzhou University, Yangzhou 225009, China; 2Joint International Research Laboratory of Agriculture and Agri-Product Safety of MOE, Yangzhou University, Yangzhou 225009, China

**Keywords:** *Vibrio parahaemolyticus*, DmeRF, cobalt homeostasis, regulation

## Abstract

Although cobalt (Co) is indispensable for life, it is toxic to cells when accumulated in excess. The DmeRF system is a well-characterized metal-response system that contributes to Co and nickel resistance in certain bacterial species. The *Vibrio parahaemolyticus* RIMD 2210633 genome also harbors a *dmeRF* operon that encodes a multiple antibiotic resistance regulator family transcriptional regulator and a cation diffusion facilitator family protein. Quantitative real-time PCR, growth curves analysis, inductively coupled plasma-mass spectrometry, β-galactosidase activity assays, electrophoretic mobility shift assays, and a mouse infection experiment were performed to characterize the function of the DmeRF system in *V. parahaemolyticus*. Zinc, copper, and Co significantly increase *dmeF* expression, with Co inducing the greatest increase. DmeF promotes *V. parahaemolyticus* growth under high-Co conditions. Additionally, increased accumulation of cellular Co in the Δ*dmeF* mutant indicates that DmeF is potentially involved in Co efflux. Moreover, DmeR represses the *dmeRF* operon by binding directly to its promoter in the absence of Co. Finally, the DmeRF system was not required for *V. parahaemolyticus* virulence in mice. Collectively, our data indicate that the DmeRF system is involved in maintaining Co homeostasis in *V. parahaemolyticus* and DmeR functioning as a repressor of the operon.

## 1. Introduction

Metals, such as cobalt (Co) and zinc (Zn), are indispensable for almost all organisms. Indeed, many proteins require metals as structural components or enzymatic cofactors [1]. Vertebrate hosts have evolved a strategy, termed nutritional immunity, to decrease the availability of metals to invading bacteria [2]. Consequently, bacteria have developed numerous and varied countermeasures such as the utilization of high-affinity metal acquisition systems [3]. Despite their essential roles, metals are harmful to bacteria when accumulated in excess [4]. Remarkably, increasing evidence shows that hosts exploit metal toxicity to defend against bacterial pathogens [5,6]. Metal homeostasis disturbance can lead to mismetalation (i.e., protein metalation with a non preferred metal) or the generation of reactive oxygen species (ROS) via the Fenton reaction; both are detrimental to cells [7]. Therefore, metal homeostasis should be tightly regulated. Since metals cannot be synthesized or degraded, bacteria maintain their homeostasis mainly through the modulation of metal import and export [4].

Co is a micronutrient that participates in various metabolic processes. Most notably, it is required by vitamin B_12_ and certain proteins [8,9]. Nonetheless, excessive amounts of Co lead to toxicity. It can cause cell damage by catalyzing the production of ROS [10]. Moreover, Co competes with iron (Fe) in a variety of metabolic processes [11,12,13]. To prevent Co toxicity, efflux systems, such as P_1B-4_-type ATPase and cation diffusion facilitator (CDF), are employed to eliminate excessive amounts of Co [14]. In *Streptococcus suis*, a P_1B-4_-type ATPase, PmtA, acts as a ferrous iron (Fe[II]) and Co efflux pump [15]. Other examples of Co-exporting P_1B-4_-type ATPases include CoaT from *Synechocystis* and CtpD from *Mycobacterium smegmatis* [16,17]. DmeF, a CDF family protein, was first identified as playing a crucial role in Co homeostasis in *Cupriavidus metallidurans* (formerly called *Wautersia metallidurans*) [18]. Subsequently, DmeF homologues have been shown to be essential for Co and nickel (Ni) resistance in certain *Rhizobiaceae* species, including *Sinorhizobium meliloti*, *Agrobacterium fabrum* (formerly called *Agrobacterium tumefaciens*), and *Rhizobium leguminosarum* [14,19,20,21]. The gene *dmeR* encodes a RcnR/CsoR family metal-responsive transcriptional regulator. Typically, *dmeF* and *dmeR* are organized in an operon (*dmeRF*) where DmeR modulates the transcription of the operon [14,19,20].

*Vibrio parahaemolyticus* (family *Vibrionaceae* and order Vibrionales) is a Gram-negative, halophilic, curved-rod-shaped bacterium that can be found in estuarine, coastal, and marine environments [22,23]. It is also a major food-borne pathogen that primarily causes acute gastroenteritis in humans. Generally, gastroenteritis caused by *V. parahaemolyticus* is associated with the ingestion of contaminated seafood [24]. Occasionally, *V. parahaemolyticus* causes wound infections (leading to cellulitis or necrotizing fasciitis) and septicemia [25]. In addition to human infections, *V. parahaemolyticus* is responsible for acute hepatopancreatic necrosis disease in shrimp, which results in considerable economic losses in shrimp aquaculture [26,27]. Whole genome sequencing of RIMD 2210633, a *V. parahaemolyticus* clinical isolate, has revealed that this bacterium uses a mechanism distinct from that of *Vibrio cholerae* to establish infection [28]. Furthermore, a locus encoding a DmeF homologue was identified in the genome.

This study aimed to characterize the function of the DmeRF system in *V. parahaemolyticus*. This system was found to play a central role in Co homeostasis. Moreover, the *dmeRF* operon is repressed by DmeR in the absence of Co.

## 2. Results

### 2.1. Identification of the DmeRF System in V. parahaemolyticus

In the genome of *V. parahaemolyticus* RIMD 2210633, the locus *VP_RS21330* is annotated as *dmeF*, whose product is the CDF family Co/Ni efflux transporter DmeF. This protein exhibits 50.72%, 49.68%, 46.15%, and 44.68% amino acid sequence identity with DmeF from *C. metallidurans*, *S. meliloti*, *A. fabrum*, and *R. leguminosarum*, respectively. The locus *VP_RS21325*, immediately adjacent to and co-transcribed with *dmeF* (Appendix A), encodes a multiple antibiotic resistance regulator (MarR) family transcriptional regulator. No significant similarity was found between this protein and DmeR from the *Rhizobiaceae* species. Despite this, *VP_RS21325* was designated as *dmeR* based on its regulatory role in *dmeF* expression (see Section 2.5). In the *Rhizobiaceae* species, *dmeR* is located upstream of *dmeF* [14,19,20,21], but in *V. parahaemolyticus*, the reverse applies (Figure 1). Multiple sequence alignment also showed that the DmeF proteins from *V. parahaemolyticus*, *C. metallidurans*, and the *Rhizobiaceae* species are highly conserved, while the DmeR protein from *V. parahaemolyticus* shows little homology with those of the *Rhizobiaceae* species (Figure 2). Together, these results revealed that the *V. parahaemolyticus* DmeF may have functions similar to its homologues from *C. metallidurans* and *Rhizobiaceae*.

### 2.2. The V. parahaemolyticus dmeF Gene Is Inducible by Zinc, Copper, and Cobalt

To evaluate the involvement of DmeF in the response of *V. parahaemolyticus* to metal toxicity, *dmeF* expression in RIMD 2,210,633 incubated with elevated levels of various metals was determined by quantitative real-time PCR (qRT-PCR) analysis. As seen in Figure 3, *dmeF* expression increased approximately 14-fold after the Co treatment compared to the H_2_O treatment. Furthermore, the Zn and Cu treatments increased *dmeF* expression by approximately 4-fold and 5-fold, respectively, while Fe(II), Mn, and Ni had no significant effect on *dmeF* expression (Figure 3).

### 2.3. DmeF Contributes to V. parahaemolyticus Growth under High-Cobalt Conditions

To explore the role of DmeRF in *V. parahaemolyticus* physiology, we generated gene deletion mutants (including the single mutants Δ*dmeR* and Δ*dmeF*, and the double mutant Δ*dmeRF*) and overexpression strains in the corresponding mutant’s background (OE*dmeR*, OE*dmeF*, and OE*dmeRF*). These strains were verified by PCR analysis and DNA sequencing.

The sensitivities of the wild type (WT), mutants, and overexpression strains to various Co concentrations were determined by growth curves analysis. In the absence of Co, all the strains displayed similar growth (Figure 4A). However, in the presence of elevated concentrations of Co, the Δ*dmeF* and Δ*dmeRF* mutants exhibited severely decreased growth compared to the WT strain (Figure 4B−D). The Δ*dmeR* mutant exhibited similar growth compared to the WT strain, while the OE*dmeF* and OE*dmeRF* strains, in which *dmeF* expression is significantly upregulated (Figure 5), grew much better than the WT strain (Figure 4B−D). Surprisingly, decreased OE*dmeR* growth was also observed in the presence of elevated Co concentrations (Figure 4B−D).

The sensitivities of the WT, Δ*dmeF*, and OE*dmeF* strains to various other metals were also evaluated. Upon supplementation of Fe(II), Mn, Zn, Cu, and Ni, Δ*dmeF* displayed almost identical growth compared to the WT strain (Appendix A). OE*dmeF* exhibited either slight or moderate growth decreases in the presence of trisodium citrate dihydrate (TCD, as a control for the Fe[II] treatment), Fe(II), Mn, Zn, and Cu, whereas it showed increased growth compared to the WT strain in the presence of Ni (Appendix A). Taken together, these results indicate that DmeF is involved in the resistance of *V. parahaemolyticus* to Co toxicity.

### 2.4. The ΔdmeF Mutant Accumulated Increased Levels of Cellular Cobalt Content

To better understand the mechanism underlying DmeF-mediated Co resistance, the WT, Δ*dmeF*, and OE*dmeF* strains grown in the presence of 0.1 mM Co were collected and analyzed for Co content using inductively coupled plasma-mass spectrometry (ICP-MS). As seen in Figure 6, the WT and Δ*dmeF* strains accumulated approximately 65 and 87 μg Co per g cells (dry weight), respectively. The OE*dmeF* strain, which had a higher expression of *dmeF*, accumulated only 6 μg Co per g cells (dry weight) (Figure 6). The cellular Co content in Δ*dmeF* was significantly higher than that in the WT and OE*dmeF* strains. Moreover, when growing in the presence of 1 mM Ni, Δ*dmeF* accumulated higher Ni content than the WT strain (although not significantly) and OE*demF* (significantly) (Appendix A).

### 2.5. DmeR Negatively Regulates the dmeRF Operon by Binding Directly to the Promoter While Cobalt Inhibits the Interaction

To test whether DmeR regulates the *dmeRF* operon, *dmeF* expression in the WT, Δ*dmeR*, and OE*dmeR* strains was determined by qRT-PCR analysis. In Δ*dmeR*, *dmeF* expression was approximately 5-fold and 31-fold higher than that of the WT and OE*dmeR* strains, respectively (Figure 7A).

The promoter region of *dmeRF* contains a predicted binding site for DmeR that consists of a DNA sequence flanked by two inverted repeats (Appendix A). The promotion of *dmeRF* was mutated by replacing the putative binding site with an unrelated DNA sequence, to generate P*_dmeRF_*’ (Appendix A). The WT and Δ*dmeR* strains harboring either pDM8 (a plasmid carrying a promoterless *lacZ* gene), P*_dmeRF_*-pDM8 (pDM8 carrying the promoter of *dmeRF*), or P*_dmeRF_*’-pDM8 (pDM8 carrying the mutant promoter of *dmeRF*) were measured for β-galactosidase activity. As shown in Figure 7B, the β-galactosidase activity produced by Δ*dmeR* harboring P*_dmeRF_*-pDM8 (2039 Miller units) was significantly higher than that produced by the WT strain harboring the same plasmid (743 Miller units). In addition, the strain (either WT or Δ*dmeR*) harboring P*_dmeRF_*’-pDM8 produced β-galactosidase activity comparable to that harboring pDM8 (Figure 7B).

To further elucidate the regulatory mechanism of DmeR, electrophoretic mobility shift assays (EMSAs) were performed. Purified recombinant DmeR (rDmeR) was incubated with the promoter probes of either *dmeRF* (WT or mutant) or *gyrB* (as a negative control) in buffers containing EDTA or Co. As seen in Figure 8A, when 0.1 mM EDTA was present in the reactions, a single shifted band became visible as the concentration of rDmeR increased. In contrast, no shifted band was observed in reactions containing 0.5 mM Co (Figure 8B). Moreover, when incubated with the mutant promoter probe of *dmeRF*, no shifted band was observed in reactions containing EDTA (Figure 8C). Regardless of the presence of EDTA or Co, rDmeR was unable to shift the negative control probes (Figure 8A–C).

Together, these results indicate that DmeR represses the *dmeRF* operon by binding directly to its promoter in the absence of Co.

### 2.6. The DmeRF System Is Not Required for V. parahaemolyticus Virulence in Mice

To investigate the role of DmeRF in *V. parahaemolyticus* virulence, an experimental infection of C57BL/6 mice was conducted. Ten mice per treatment were intraperitoneally infected with either phosphate-buffered saline (PBS) or one of four *V. parahaemolyticus* strains: the WT, Δ*dmeR*, Δ*dmeF*, or Δ*dmeRF* strains. At 12 h post infection, the survival rates for mice in the WT, Δ*dmeR*, Δ*dmeF*, and Δ*dmeRF* groups were 40%, 50%, 20%, and 20%, respectively (Figure 9). The remaining mice in the WT and Δ*dmeRF* groups died during the following 12 h, and 10% of the mice infected with the Δ*dmeR* or Δ*dmeF* strains survived over the course of the experiment (Figure 9). In contrast, all mice injected with PBS survived. These results suggest that the DmeRF system has no significant role in *V. parahaemolyticus* virulence in mice.

## 3. Discussion

Even though the maintenance of metal homeostasis is essential for bacterial physiology and pathogenesis, not much is known about the mechanisms by which *V. parahaemolyticus* responds to metal overload. Metal efflux is one of the most important mechanisms employed by bacteria to reduce the damage caused by metal influxes [4]. The DmeRF system is a well-characterized metal-response system that is involved in Co resistance in *C. metallidurans* and Co/Ni resistance in certain *Rhizobiaceae* species [14,18,19,20,21]. The DmeRF system was demonstrated to contribute to the maintenance of Co homeostasis in *V. parahaemolyticus* by providing the following lines of evidence: (i) *V. parahaemolyticus* DmeF shares a high level of homology (approximately 44% to 51% amino acid sequence identity) with its homologues, all of which are involved in Co (and Ni) resistance; (ii) *dmeF* expression is significantly upregulated in response to Zn, Cu, and Co exposure; (iii) the *dmeF* deletion mutants exhibit increased sensitivity to Co stress, while the overexpression strains that have higher *dmeF* expression exhibit decreased sensitivity to Co stress; (iv) when cultured in a medium supplemented with Co, the cellular Co content in the Δ*dmeF* mutant is significantly higher than that in the WT and OE*dmeF* strains; and (v) in the presence of Co, DmeR dissociates from the *dmeRF* promoter, allowing the transcription of *dmeF*.

The role of the DmeRF system in the metal tolerances of *C. metallidurans* and certain *Rhizobiaceae* species is well studied [14,18,19,20,21]. Therefore, BlastP analyses and multiple sequence alignments were performed to examine the level of homology exhibited between the DmeRF systems of *V. parahaemolyticus* and their homologues in these other species. While the DmeF proteins share high levels of homology, no significant similarity was observed between the DmeR of *V. parahaemolyticus* and those of the *Rhizobiaceae* species. It is not surprising as *V. parahaemolyticus* DmeR is a MarR family regulator while the others belong to the RcnR/CsoR family. Moreover, the organization of the *V. parahaemolyticus demRF* operon is different from those of the *Rhizobiaceae* species. Consequently, it is worthwhile to explore the role of DmeRF in *V. parahaemolyticus*.

Typically, bacteria respond to metal excess by expressing specific genes. Therefore, *dmeF* expression in the presence of elevated levels of various metals was measured using qRT-PCR analysis. *dmeF* expression is induced by Zn, Cu, and Co, with Co serving as the most potent inducer. The expression of *dmeF* differs among *V. parahaemolyticus*, *C. metallidurans*, and the *Rhizobiaceae* species. In *C. metallidurans*, *dmeF* expression is constitutive and cannot be induced by metals [18]. By contrast, *dmeF* expression is strongly induced by Co, Ni, and Cu in *S. meliloti* [19], and specifically induced by Co and Ni in *A. fabrum* and *R. leguminosarum*, with Co being a more potent inducer [14,20]. Although Cu induces *demF* expression in *S. meliloti*, the Δ*dmeF* mutant exhibited no difference in growth compared with the WT strain under high-Cu conditions [19]. Similarly, in *V. parahaemolyticus*, there was no major difference in the growth of the Δ*dmeF* and WT strains under high-Zn or -Cu conditions. A similar observation has also been made for the Fe(II) and Co efflux pump PmtA in *S. suis*. While *pmtA* expression is induced by Fe(II), Co, and Ni, the Δ*pmtA* mutant displayed no growth decrease under Ni stress [15]. We speculate that excessive amounts of Zn or Cu may change DmeR conformation, resulting in partial derepression of the *dmeRF* operon.

In line with the induction of *dmeF* expression by Co, the *dmeF* deletion mutants exhibited obvious growth inhibition under high-Co conditions, whereas, the *dmeF* overexpression strains grew better than the WT strain in the presence of elevated Co levels. These results clearly suggest that DmeF is involved in Co resistance in *V. parahaemolyticus*, which is consistent with the observations in *C. metallidurans* and certain *Rhizobiaceae* species [14,18,19,20,21]. In *A. fabrum*, the Δ*dmeF* mutant accumulated significantly higher levels of cellular Co content than the WT strain when cultured in a Co-rich medium [14]. A similar observation was made in our study, indicating that DmeF potentially mediates Co resistance by Co efflux. Yet, growth of the Δ*dmeF* and Δ*dmeRF* strains was not completely inhibited by high concentrations of Co, suggesting that additional Co resistance systems likely exist in *V. parahaemolyticus*. Surprisingly, the OE*dmeR* strain, which showed very low *dmeF* expression, exhibited more severe growth inhibition than Δ*dmeF* and Δ*dmeRF* under Co stress. We speculate that DmeR may play a role in the regulation of other Co resistance systems. Interestingly, the OE*dmeF* strain exhibited increased growth compared to the WT strain in the presence of high concentrations of Ni. Consistent with their growth, OE*dmeF* accumulated significantly lower levels of cellular Ni content under Ni conditions. These results indicate that in *V. parahaemolyticus*, DmeF may be involved in Ni resistance, albeit in a less prominent role than in Co resistance.

qRT-PCR analysis, β-galactosidase activity assays, and EMSAs showed that DmeR, a MarR family regulator, represses the *dmeRF* operon by binding directly to the promoter in the absence of Co. In several studied *Rhizobiaceae* species, *dmeF* is co-transcribed with a gene encoding the RcnR/CsoR family regulator, whose repression of *dmeRF* transcription has been confirmed in *A. fabrum* [14,19,20]. Accordingly, we speculate that *V. parahaemolyticus* and *Rhizobiaceae* adopt regulators belonging to different families to modulate the conserved Co resistance system. Furthermore, the mechanisms of the DmeRF system in *V. parahaemolyticus* were proposed according to our results and the findings in *A. fabrum* [14]. In the presence of limited or normal concentrations of Co, DmeR binds to the promoter region of *dmeRF*, resulting in transcriptional repression, whereas under high-Co conditions, excessive amounts of Co result in DmeR dissociating from the promoter, probably by causing a conformational change, and consequently, the repression is relieved.

In *A. fabrum*, the inactivation of *dmeF* has no effect on bacterial virulence in *Nicotiana benthamiana* [14]. Likewise, the DmeRF system plays no significant role in *V. parahaemolyticus* virulence in mice. We speculate that the Co concentration is low in mouse tissues, hence the DmeRF system does not serve an important function during the infectious process.

In conclusion, a metal-response system, DmeRF, composed of the metal efflux pump DmeF and the regulator DmeR, has been identified and characterized in *V. parahaemolyticus*. The DmeRF system contributes to the maintenance of Co homeostasis in *V. parahaemolyticus*, and DmeR functions as a transcriptional repressor of the *dmeRF* operon in the absence of Co.

## 4. Materials and Methods

### 4.1. Bacterial Strains, Culture Conditions, Plasmids, and Primers

The bacterial strains and plasmids used in this study are listed in Table 1. All *Escherichia coli* strains, *V. parahaemolyticus* RIMD 2210633, and its derivatives were routinely grown at 37 °C in Luria-Bertani (LB) broth or on LB agar. When required, carbenicillin, chloramphenicol, and isopropyl β-D-1-thiogalactopyranoside (IPTG) were supplemented at 50 μg/mL, 25 μg/mL, and 1 mM, respectively. The primers used in this study are listed in Table 2.

### 4.2. RNA Extraction and qRT-PCR Analysis

An overnight culture of the RIMD 2,210,633 strain was diluted 1:100 in LB broth and grown to the early-exponential phase (OD_600_ of ~0.7). Seven 1 mL aliquots were removed, 2 μL of H_2_O was added to one and metal solutions were added to the rest, one to each, to create final concentrations of 1 mM FeSO_4_, 1 mM MnSO_4_, 0.5 mM ZnSO_4_, 1 mM CuSO_4_, 0.25 mM CoSO_4_, and 1 mM NiSO_4_. After further incubation at 37 °C for 15 min, bacterial cells were collected by centrifugation. Total RNA was isolated from the cell pellets using an Eastep Super Total RNA Isolation Kit (Promega, Shanghai, China). Three independent experiments were performed to obtain triplicate biological samples.

In another experiment, overnight cultures of the WT, Δ*dmeR*, and OE*dmeR* were separately diluted 1:100 in LB broth and grown to the early-exponential phase (OD_600_ of ~0.7). Then, bacterial cells were collected for RNA isolation as described above. Three independent experiments were performed to obtain triplicate biological samples.

After evaluations of RNA integrity and measurements of RNA concentrations, the qualified RNAs were subjected to qRT-PCR analysis. cDNA was generated from approximately 200 ng of RNA per sample using ToloScript RT EasyMix for qPCR (with 2-step gDNA Erase-Out) (TOLOBIO, Shanghai, China). Quantitative PCR was performed on the StepOnePlus Real-Time PCR System (Applied Biosystems, Waltham, MA, USA) using NovoStart SYBR qPCR SuperMix Plus (Novoprotein, Shanghai, China) and gene-specific primers (Table 2). Gene expression levels were analyzed using the 2^− ΔΔCT^ method [32], with *gyrB* as the internal standard.

### 4.3. Construction of Gene Deletion and Overexpression Strains

The gene deletion mutants Δ*dmeR*, Δ*dmeF*, and Δ*dmeRF* were generated via allelic exchange using the pDM4 plasmid [29], as previously described [33]. The overexpression strains OE*dmeR*, OE*dmeF*, and OE*dmeRF* were constructed using the pMMB207 plasmid [30], as previously described [33].

### 4.4. Growth Evaluation

The WT, Δ*dmeR*, Δ*dmeF*, Δ*dmeRF*, OE*dmeR*, OE*dmeF*, and OE*dmeRF* strains were grown to the mid-exponential phase (OD_600_ of ~2) and diluted 1:100 in LB broth Supplemented with one of four concentrations of CoSO_4_ (0, 0.1, 0.2, or 0.3 mM) or specific high concentrations of another metal (2 mM FeSO_4_, 1 mM MnSO_4_, 1.25 mM ZnSO_4_, 2 mM CuSO_4_, or 1 mM NiSO_4_). To alleviate Fe(II) oxidation, 1 g/L of TCD was supplemented to the medium containing FeSO_4_ [34]. The cultures were transferred into 96-well plates with 200 μL per well and three wells per treatment. The plates were incubated in a shaker at 37 °C and 120 rpm, and the OD_595_ values were measured hourly using a CMax Plus plate reader (Molecular Devices, San Jose, CA, USA).

### 4.5. Intracellular Metal Content Analysis

The WT, Δ*dmeF*, and OE*dmeF* strains were grown to the mid-exponential phase (OD_600_ of ~2) and diluted 1:100 in LB broth supplemented with either 0.1 mM CoSO_4_ or 1 mM NiSO_4_. After incubation for another 6 h, bacterial cells were collected by centrifugation. Five independent experiments were performed to obtain five biological samples for each strain. Sample washing, drying, digestion, and dilution were performed as previously described [35]. Co/Ni content in these samples was analyzed by ICP-MS at Yangzhou University. The metal content was expressed as μg of Co/Ni per g of cells (dry weight).

### 4.6. Construction of LacZ Fusion Strains and β-galactosidase Activity Assays

The promoter of the *dmeRF* operon was mutated by replacing the putative binding site for DmeR with an unrelated DNA sequence. The DNA carrying the mutant promoter of *dmeRF* (P*_dmeRF_*’) was synthesized by Tsingke Biotechnology Co., Ltd. (Beijing, China).

The promoter of *dmeRF* (either WT or mutant) was cloned into pDM8, a plasmid carrying a promoterless *lacZ* gene [31]. The resulting plasmid P*_dmeRF_*-pDM8/P*_dmeRF_*’-pDM8 was transformed into *E. coli* S17-1 λpir and then conjugated into the WT strain and Δ*dmeR*. The WT strain and Δ*dmeR* harboring the empty pDM8 plasmid served as the control strains.

β-galactosidase activity assays were performed as previously described [36,37], with some modifications. Overnight cultures of the LacZ fusion strains were diluted 1:100 in LB broth and grown to the early-exponential phase (OD_600_ of ~0.7). Bacterial cells were collected from 1 mL of each culture by centrifugation. The cell pellets were resuspended in 1 mL PM buffer (60 mM Na_2_HPO_4_, 40 mM NaH_2_PO_4_, 10 mM KCl, 1 mM MgSO_4_, and 50 mM β-mercaptoethanol, pH 7.0). The A_600_ values of the bacterial suspensions were measured. For each suspension, 200 μL of suspension, 30 μL of chloroform, and 30 μL of 0.1% SDS were added to 500 μL of PM buffer, and then vortexed vigorously to lyse bacterial cells. The reaction was started by adding 200 μL of o-nitrophenyl-β-galactopyranoside (4 mg/mL in PM buffer). When the mixture turned yellowish, the reaction was stopped by adding 400 μL of 1 M Na_2_CO_3_. The mixture was centrifuged, and the A_420_ value of the supernatant was measured. β-galactosidase activity, in Miller units, was calculated as A_420_ × 1000 × min^−1^ × mL^−1^ × A_600_^−1^.

### 4.7. rDmeR Expression, Purification, and EMSAs

The *dmeR* gene was cloned into the pET-30a plasmid, and the resulting plasmid, pET30a-*dmeR*, was transformed into *E. coli* BL21(DE3). The strain was grown at 37 °C to the mid-exponential phase (OD_600_ of ~0.8). Then, 0.5 mM of isopropyl-β-D-thiogalactopyranoside (IPTG) was added to induce DmeR expression. After further growth at 28 °C for 4 h, bacterial cells were harvested by centrifugation. The cell pellets were resuspended in binding buffer (20 mM Tris-HCl, 500 mM NaCl, and 20 mM imidazole, pH 8.0) and lysed by sonication. rDmeR purification and assessment of its quality and concentration were performed as previously described [38].

EMSAs were performed as previously described [38]. DNA fragments containing the promoters of the *dmeRF* operon (WT or mutant) or *gyrB* (negative control) were PCR amplified and then purified. The 20 μL reaction mixtures contained 20 ng of the DNA fragments, varying amounts of rDmeR (0, 0.1, 0.2, 0.5, or 1.0 μg), and 200 ng of poly (dI:dC) in either EMSA buffer 1 (150 mM NaCl, 0.1 mM DTT, 0.1 mM EDTA, and 10 mM Tris, pH 7.4) or the Co containing EMSA buffer 2 (150 mM NaCl, 0.1 mM DTT, 0.5 mM CoSO_4_, and 10 mM Tris, pH 7.4). The mixtures were incubated at 25 °C for 30 min and then resolved by 6% native polyacrylamide gel electrophoresis in 0.5 × TBE buffer at 100 V for 2 h. After staining with SYBR Green I for 30 min, the gel was photographed.

### 4.8. Mouse Infection Experiment

A total of 50 female C57BL/6 mice (specific, pathogen-free, 8−12 weeks old) were randomly divided into five groups (10 mice per group). The WT, Δ*dmeR*, Δ*dmeF*, and Δ*dmeRF* strains were grown in LB broth at 30 °C for 12 h and adjusted to 1 × 10^9^ CFU/mL in PBS. For groups I to IV, the mice were intraperitoneally infected with 100 μL of the corresponding strain. Mice in group V were injected with 100 μL of PBS and served as the control. Mouse survival was recorded twice daily for seven days.

### 4.9. Bioinformatic and Statistical Analysis

BlastP was performed to analyze the amino acid sequence identities of homologous proteins. Clustal Omega (https://www.ebi.ac.uk/Tools/msa/clustalo/ (accessed on 12 October 2022)) was used for multiple sequence alignments; the results were visualized using ESPript 3.0 [39]. Promoters were predicted using BPROM (http://linux1.softberry.com/berry.phtml (accessed on 12 October 2022)).

GraphPad Prism 5 (San Diego, CA, USA) was used for statistical analysis. Gene expression, metal content, and β-galactosidase activity were analyzed by one-way analysis of variance with a Bonferroni’s post-test. The log-rank test was used for analyzing mouse survival curves.

## Figures and Tables

**Figure 1 ijms-24-00414-f001:**
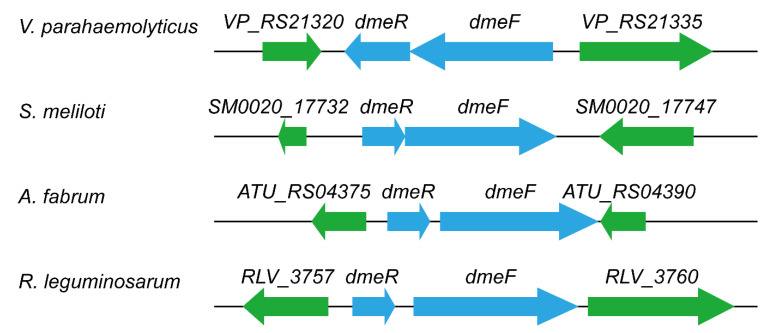
Genetic structures of the *dmeRF* systems in *V. parahaemolyticus* and several *Rhizobiaceae* species. Arrows indicate the direction of transcription. The structure analyses were performed using the following genomes: *V. parahaemolyticus* RIMD 2210633, NC_004605.1; *S. meliloti* CCNWSX0020, AGVV01000035.1; *A. fabrum* C58, NC_003062.2; and *R. leguminosarum* UPM791, CP025509.1.

**Figure 2 ijms-24-00414-f002:**
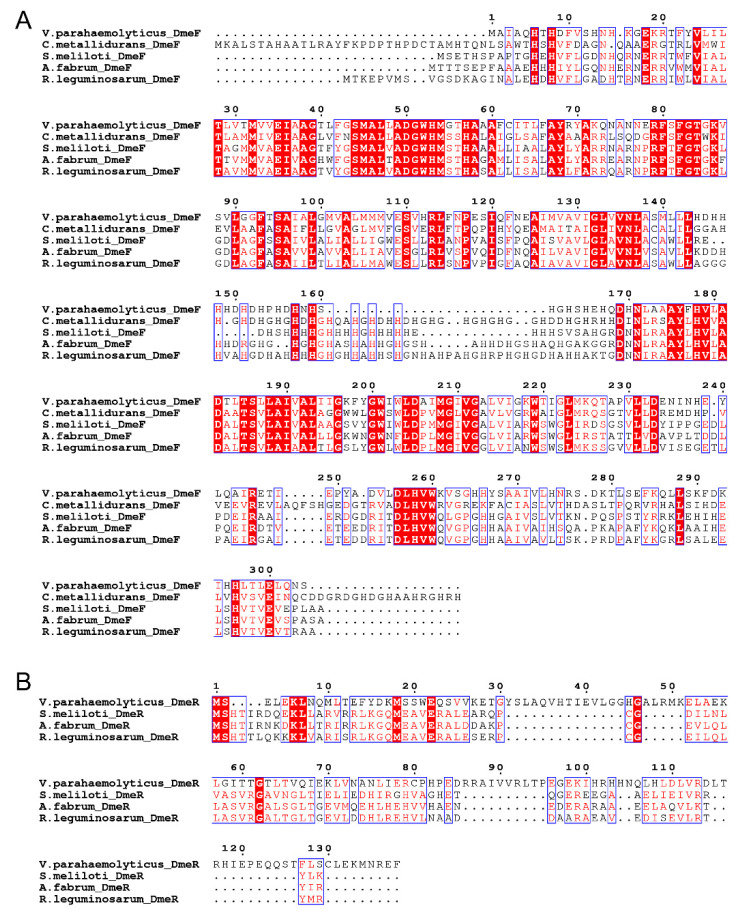
Multiple sequence alignments of the DmeF (**A**) and DmeR (**B**) homologues. Identical residues are in white letters on a red background, and similar residues are in red letters on a white background. The GenBank accession numbers are as follows: *V. parahaemolyticus* DmeF, WP_005477256.1; *C. metallidurans* DmeF, ABF07084.1; *S. meliloti* DmeF, EHK76549.1; *A. fabrum* DmeF, AAK86697.2; *R. leguminosarum* DmeF, AVC48922.1; *V. parahaemolyticus* DmeR, WP_005463375.1; *S. meliloti* DmeR, EHK76548.1; *A. fabrum* DmeR, AAK86696.2; and *R. leguminosarum* DmeR, AVC48921.1. DmeR homologue has not been described in *C. metallidurans*; thus, the analysis does not include this protein.

**Figure 3 ijms-24-00414-f003:**
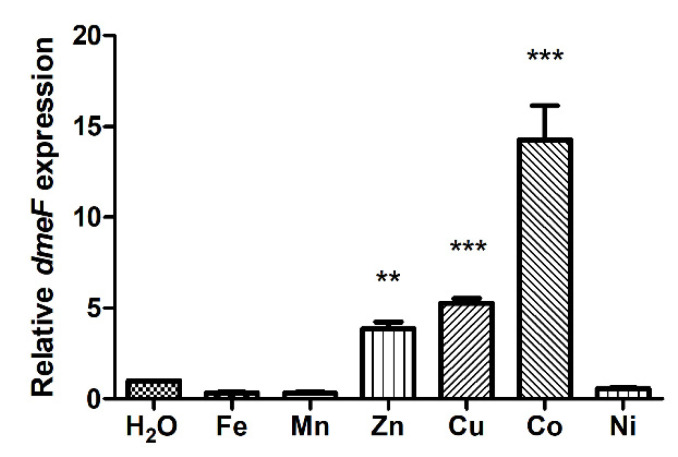
Induction of *dmeF* expression by various metals. *V. parahaemolyticus* RIMD 2,210,633 was incubated in H_2_O, 1 mM FeSO_4_, 1 mM MnSO_4_, 0.5 mM ZnSO_4_, 1 mM CuSO_4_, 0.25 mM CoSO_4_, or 1 mM NiSO_4_ for 15 min. Expression of *dmeF* is reported relative to the H_2_O treatment. Results represent the means and standard deviations (SD) from three biological replicates. The data were analyzed using one-way analysis of variance along with Bonferroni’s post-test. **, *p* < 0.01; ***, *p* < 0.001.

**Figure 4 ijms-24-00414-f004:**
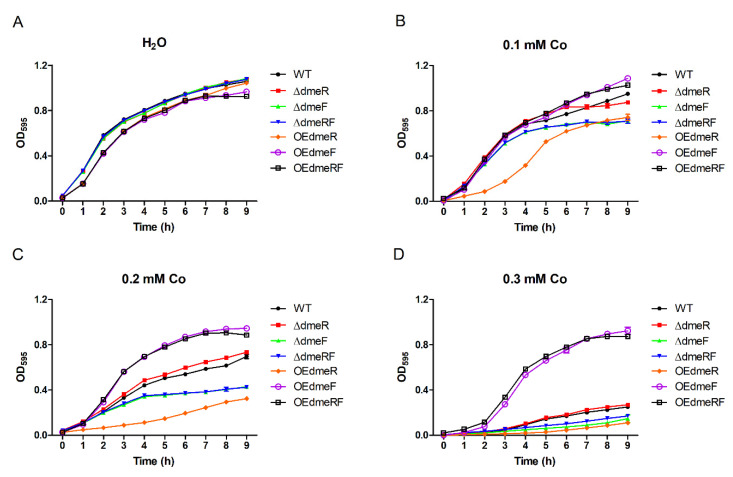
Growth curves of the *V. parahaemolyticus* strains grown in the absence of Co (**A**) or in 0.1 mM (**B**), 0.2 mM (**C**), or 0.3 mM (**D**) solutions of Co. The experiments were performed at least three times; the results represent the means and SD from three wells in a representative experiment.

**Figure 5 ijms-24-00414-f005:**
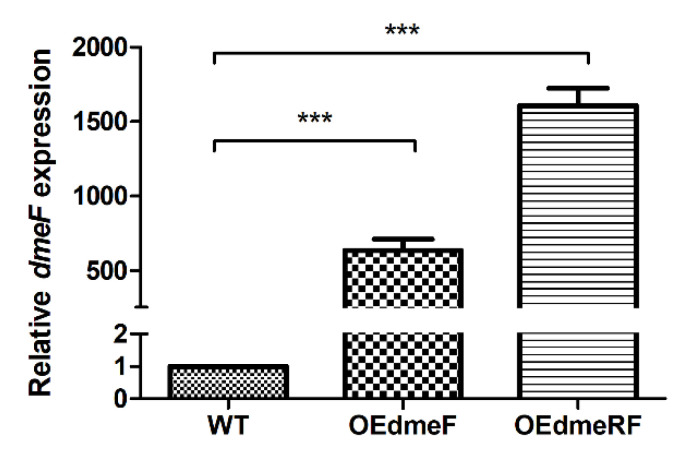
qRT-PCR analysis of *dmeF* expression in the WT, OE*dmeF*, and OE*dmeRF* strains. Results represent the means and SD from three biological replicates. The data were analyzed using one-way analysis of variance along with Bonferroni’s post-test. ***, *p* < 0.001.

**Figure 6 ijms-24-00414-f006:**
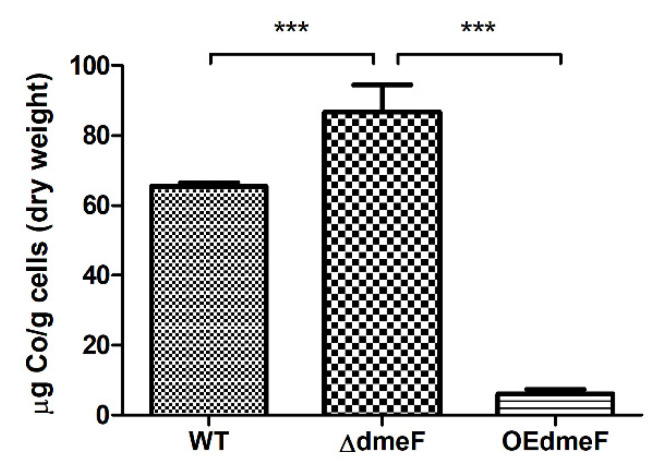
Cellular Co content in the WT, Δ*dmeF*, and OE*dmeF* strains. These strains were grown in the presence of 0.1 mM CoSO_4_ for 6 h. Cellular Co content was analyzed by inductively coupled plasma-mass spectrometry. Results represent the means and SD from five biological replicates. The data were analyzed using one-way analysis of variance along with Bonferroni’s post-test. ***, *p* < 0.001.

**Figure 7 ijms-24-00414-f007:**
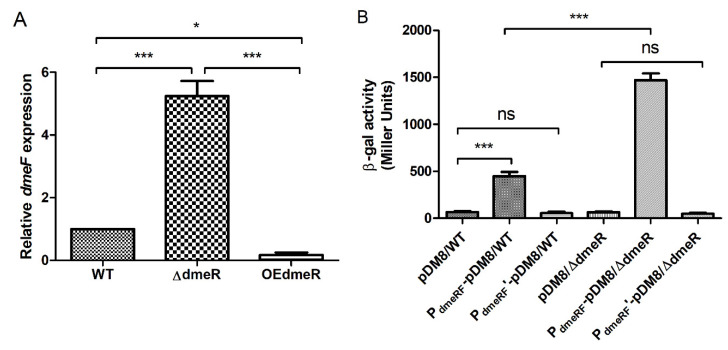
DmeR regulates the *dmeRF* operon. (**A**) qRT-PCR analysis of *dmeF* expression in the WT, Δ*dmeR*, and OE*dmeR* strains. Results represent the means and SD from three biological replicates. (**B**) Assays measuring the β-galactosidase activities of early-exponential phase cells of the WT and Δ*dmeR* strains harboring either pDM8 (a plasmid carrying a promoterless *lacZ* gene), P*_dmeRF_*-pDM8 (pDM8 carrying the promoter of *dmeRF*), or P*_dmeRF_*’-pDM8 (pDM8 carrying the mutant promoter of *dmeRF*). Results represent the means and SD from three independent experiments performed in duplicate. The data were analyzed using one-way analysis of variance along with Bonferroni’s post-test. ns, not significant; *, *p* < 0.05; ***, *p* < 0.001.

**Figure 8 ijms-24-00414-f008:**
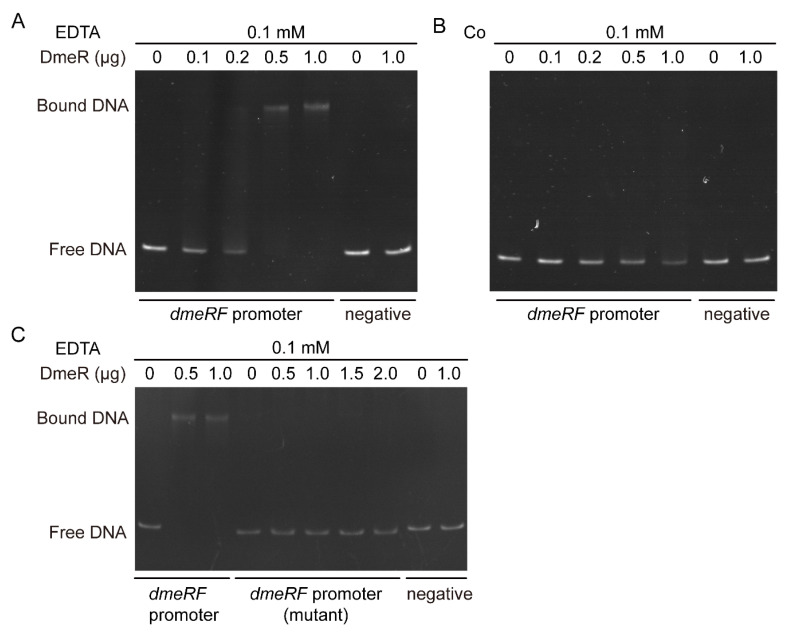
Electrophoretic mobility shift assays. (**A**,**B**) Purified rDmeR was incubated with the promoter probes of either *dmeRF* or *gyrB* (negative control) in buffers containing EDTA (**A**) or Co (**B**). (**C**) Purified rDmeR was incubated with the promoter probes in buffers containing EDTA. The mutant *demRF* promoter probe was generated by replacing the putative binding site for DmeR with an unrelated DNA sequence. The rDmeR was added to each reaction mixture in the amounts indicated. The images are representative of at least three independent experiments.

**Figure 9 ijms-24-00414-f009:**
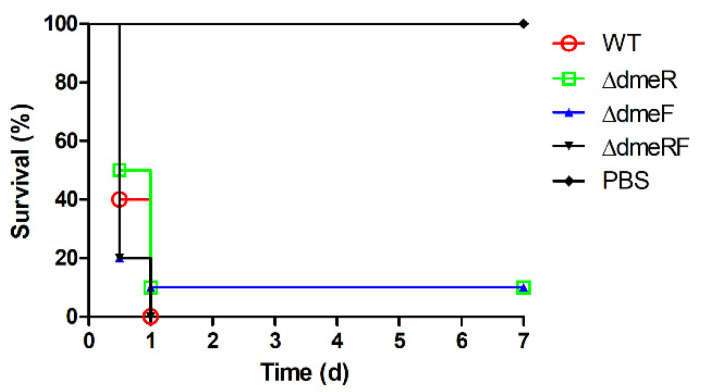
Survival curves of mice infected with one of four *V. parahaemolyticus* strains or PBS. Ten mice per treatment were intraperitoneally infected with 1 × 10^8^ CFU of the WT, Δ*dmeR*, Δ*dmeF*, or Δ*dmeRF* strains, or injected with 100 μL of PBS as the control. The data were analyzed using the log-rank test.

**Table 1 ijms-24-00414-t001:** Bacterial strains and plasmids used in this study.

Strain or Plasmid	Relevant Characteristics ^1^	Source or Reference
Strains		
*E. coli*		
DH5α λpir	Cloning host for pDM4, pMMB207, and pDM8	Laboratory collection
S17-1 λpir	Conjugal donor for pDM4, pMMB207, and pDM8	Laboratory collection
DH5α	Cloning host for pET-30a	Laboratory collection
BL21(DE3)	Expression host for pET-30a	Laboratory collection
*V. parahaemolyticus*		
RIMD 2,210,633 (WT)	Clinical isolate, Carb^R^	[28]
Δ*dmeR*	*dmeR* deletion mutant of RIMD 2210633	This study
Δ*dmeF*	*dmeF* deletion mutant of RIMD 2210633	This study
Δ*dmeRF*	*dmeR* and *dmeF* double mutant of RIMD 2210633	This study
OE*dmeR*	*dmeR* overexpression strain in Δ*dmeR* background	This study
OE*dmeF*	*dmeF* overexpression strain in Δ*dmeF* background	This study
OE*dmeRF*	*dmeRF* overexpression strain in Δ*dmeRF* background	This study
pDM8/WT	RIMD 2,210,633 harboring pDM8	This study
P*_dmeRF_*-pDM8/WT	RIMD 2,210,633 harboring P*_dmeRF_*-pDM8	This study
P*_dmeRF_*’-pDM8/WT	RIMD 2,210,633 harboring P*_dmeRF_*’-pDM8	This study
pDM8/Δ*dmeR*	Δ*dmeR* harboring pDM8	This study
P*_dmeRF_*-pDM8/Δ*dmeR*	Δ*dmeR* harboring P*_dmeRF_*-pDM8	This study
P*_dmeRF_*’-pDM8/Δ*dmeR*	Δ*dmeR* harboring P*_dmeRF_*’-pDM8	This study
Plasmids		
pDM4	Suicide vector containing a *sacB* counterselectable marker; Cm^R^	[29]
pDM4-Δ*dmeR*	Knockout vector for *dmeR* deletion	This study
pDM4-Δ*dmeF*	Knockout vector for *dmeF* deletion	This study
pDM4-Δ*dmeRF*	Knockout vector for *dmeRF* deletion	This study
pMMB207	Wide-host-range low-copy-number vector; Cm^R^	[30]
pMMB207-*dmeR*	pMMB207 containing *dmeR* and an additional ribosome-binding site	This study
pMMB207-*dmeF*	pMMB207 containing *dmeF* and an additional ribosome-binding site	This study
pMMB207-*dmeRF*	pMMB207 containing *dmeRF* and an additional ribosome-binding site	This study
pDM8	Plasmid containing the promoterless lacZ gene; Cm^R^	[31]
P*_dmeRF_*-pDM8	pDM8 containing the promoter of *dmeRF*	This study
P*_dmeRF_*’-pDM8	pDM8 containing the mutant promoter of *dmeRF*	This study
pET-30a	Expression vector; Kan^R^	Novagen
pET30a-*dmeR*	pET-30a containing *dmeR*	This study

^1^ Carb^R^, carbenicillin resistant; Cm^R^, chloramphenicol resistant; Kan^R^, kanamycin resistant.

**Table 2 ijms-24-00414-t002:** Primers used in this study.

Primer	Sequence (5′-3′) ^1^	Size (bp)	Target Gene
Q*dmeF*-F	CACCAAAGCACCAACGATCC	151	An internal region of *dmeF*
Q*dmeF*-R	CGAGCACCAGGACCACAATC		
Q*gyrB*-F	GGTGGTATTCAAGCGTTCGTTC	116	An internal region of *gyrB*
Q*gyrB*-R	TGCATTGCCACTTCTACCGAG		
*dmeR*-LA-F	TCCCCCGGGCCACCACAAACGCTCTCTG	738	The left arm of *dmeR*
*dmeR*-LA-R	GATGAGCGAACCGGGAATTCTAGGTTTCA		
*dmeR*-RA-F	ATTCCCGGTTCGCTCATCGTCAGCTATTT	732	The right arm of *dmeR*
*dmeR*-RA-R	CCGCTCGAGTGTTCGCTTATCGCTATGCT		
*dmeR*-in-F	AAAGGTTGATTGCTGCTCTG	325	An internal region of *dmeR*
*dmeR*-in-R	TTCTTGGGAACAGTCAGTCG		
*dmeR*-out-F	GCATTTTGTGTTGGTGTGACT	338/733	A fragment containing *dmeR*
*dmeR*-out-R	CGATTGAGCCATACGCAG		
OE*dmeR*-F	TCCCCCGGGTAAGGAGGTAGGATAATAATGAGCGAACTAGAAAAGTTGAA	417	*dmeR* and an additional ribosome-binding site
OE*dmeR*-R	AACTGCAGCTAGAATTCCCGGTTCATTTT		
*dmeF*-LA-F	TCCCCCGGGATTATGCTTGCCACCGCT	714	The left arm of *dmeF*
*dmeF*-LA-R	CACGCACGATAAATAGCTGACGATGAGCGA		
*dmeF*-RA-F	TCAGCTATTTATCGTGCGTGTGTTGTGC	723	The right arm of *dmeF*
*dmeF*-RA-R	CCGCTCGAGAGCACGCCATTACGATAGAG		
OE*dmeF*-F	TCCCCCGGGTAAGGAGGTAGGATAATAATGGCAATAGCACAACACAC	929	*dmeF* and an additional ribosome-binding site
OE*dmeF*-R	AACTGCAGTAGTTCGCTCATCGTCAGC		
*dmeF*-in-F	CATCTAACCAAATCCATCCG	291	An internal region of *dmeF*
*dmeF*-in-R	AATCCGTTCACCGTTTGTT		
*dmeF*-out-F	CAGCACTTCAATGGTATGGAC	279/1157	A fragment containing *dmeF*
*dmeF*-out-R	AGTGATGAACGCCTTTCTTAGT		
*dmeR*-LA-R2	ACGCACGATACCGGGAATTCTAGGTTTCA		
*dmeF*-RA-F2	ATTCCCGGTATCGTGCGTGTGTTGTGC		
OE*dmeRF*-F	TCCCCCGGGTAAGGAGGTAGGATAATAATGGCAATAGCACAACACAC	1344	*dmeRF* and an additional ribosome-binding site
OE*dmeRF*-R	AACTGCAGCGGATGAAACCTAGAATTCC		
P*dmeRF*-F	CGGATCCGGGGAATTCCCGGGTAAGCGGCTGATTCCCAAAC	208	The promoter of *dmeRF*, for β-galactosidase activity assays
P*dmeRF*-R	AAGCTTATCGATTCGCCCGGGGCGTGTGTTGTGCTATTGCC		
*dmeR*-F	CGCGGATCCATGAGCGAACTAGAAAAGTTGAA	417	The *dmeR* gene
*dmeR*-R	CCGCTCGAGCTAG AATTCCCGGTTCATTTT		
P*dmeRF*-F2	TAAGCGGCTGATTCCCAAAC	208	The promoter of *dmeRF*, for EMSAs
P*dmeRF*-R2	GCGTGTGTTGTGCTATTGCC		
P*gyrB*-F	CAAGGGCAACATCTTACAGC	215	The promoter of *gyrB*
P*gyrB*-R	TCTATCCTGCCATGTTCCAC		

^1^ The underlined sequences are restriction sites.

## Data Availability

Not applicable.

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
