# Peer review of "The DmeRF System Is Involved in Maintaining Cobalt Homeostasis in Vibrio parahaemolyticus"

_ijms, 2022, doi:10.3390/ijms24010414_

Round 1

Reviewer 1 Report

The authors want to know how the DmeRF system contributes to metal homeostasis and virulence in this human pathogen.

DmeR is a MarR family regulator, while previous examples are RcnR/CsoR type; thus there is some uncertainty about demRF function.

They provide some evidence that DmeRF is involved primarily in Co homeostasis, based on changes in metal sensitivity and accumulation in mutants.

The genetic data are consistent with DmeR acting as a transcriptional repressor.

This is a well written and well presented MS, although less substantial than ideal.

fig 4 could be supplemental data, and would be more meaningful with a diagram of the location of the primers and expected products in the wt and mutant loci. its not immediately clear what is expected and how the results support the expected locus structure.

"complemented" DmeR strain (and other strains) are not actually complemented (i.e. phenotype restored to wild-type)

appears that "complemented strain is actually o/e strain - which is not ideal. need to add complemented strain with same expression as normal, or clarify that the "complemented" strain is an overexpression strain instead.

On a related point, the supplemental expression data showing the effect of these manipulations (S2) should be in the main paper.

The metal tolerance experiments shown in the supplemental data  are problematic - they show clear effects of these genetic manipulations on zinc, cu and ni tolerance, but these effects are not really addressed by the authors.  In particular the effect of the genetic changes (mutation/ overexpression) on Ni sensitivity clearly suggests a role for demRF in Ni homeostasis, but this is deemphasized by the authors. That data should be in the main paper, not in a supplement. In addition, some data for effect of these changes on Ni accumulation would be useful. ICP-MS analysis would reveal the full extent of these changes on elemental content and better support the authors arguments for specificity. As it is their data is compatible with a broad substrate specificity/regulatory response (fig 3) for this operon, but the authors seem focused on Co.

increased Co content does not prove that DmeF works via efflux - it could regulate influx by another factor for example. The authors should demonstrate their contention directly with efflux assays (loading cells with isotope to measure the rate of efflux, or by exchange with unlabeled Co).

In fig 7, authors use a different promoter fragment (gyrB) as a negative control, but the appropriate negative control is the demRF fragment with a scrambled demR binding site, to demonstrate specificity of binding.

do the authors know what the binding site is? If not, they should perform a deletion analysis.

Likewise, showing mutation of the binding site eliminates demR function in vivo would be required support for their model (fig 9).

If one of the main questions to address was what effect this operon has on infectivity, I dont think the authors sufficiently tested this idea.

The authors don't report any role of this operon in infection, but the only output considered was death. perhaps a more detailed examination of the mice would reveal more subtle differences (for example, in bacterial load within infected animals). 

It also seems as though the applied bacterial load was extreme, given the rapid death of the animals (all within a day) - perhaps an effect might have been seen if a lower inoculum were used, so that the mice were less severely affected.

minor points- overall the writing is very clear and error free, but there are some typos

dont really need the model figure (9)- its simple, and takes up space.

some supplemental figures are very low resolution and combined with the small data points are difficult to understand. better use of color and data point size is recommended.

Reviewer 2 Report

The manuscript entitled "The DmeRF System is Involved in Maintaining Cobalt Homeostasis in Vibrio parahaemolyticus" describes the function of this operon in V. parahaemolyticus. Overall, this is a very straightforward report that is easy to read and understand. The experiments are designed appropriately and yield largely unquestionable data. The results are in line with what has been observed for DmeRF in other organisms. 

My only significant issue with this manuscript is in regard to the affect of Zn, Cu and Ni. Zn and Cu induce expression of DmeF (Figure 3) and the mutants appear to grow better than the WT under conditions of Zn and Cu stress (Figure S3), despite the claim that their growth is "almost identical" to WT. A real statistical analysis would be required to validate this claim that is repeated in the Discussion. The authors acknowledge that the complemented (overexpressing) strains grow better than WT in high Ni conditions. 

I would be interested to see some of the other techniques in this manuscript applied to the effect of Zn, Cu and Ni. In particular, metal accumulation assays as in Figure 6 would inform on whether DmeF can export these other metals, especially Ni since it seems to confer resistance. The EMSA assay in the presence of these metal would also be interesting, especially since the authors theorize in the discussion that DmeR may bind these other metals weakly and derepress the operon, as is suggested by RT-PCR data. This is especially interesting since the DmeR is of a different family (MarR) in V. parahaemolyticus than in other organisms, and it is unknown how specific its metal binding properties are. 

Minor Issues: The supplementary figure 3 is too small and very difficult to read. I would also suggest inverting Figure 7C and D and possibly adjusting the contrast to make bands easier to see. The legend to figure 4 should specify the meaning of "In" and "Out". I assume this refers to the annealing site of primers being within or outside of the target gene, but that is not immediately clear. An indication of the predicted sizes of bands might also be nice. 

Reviewer 3 Report

First of all, I would like to congratulate the authors of this article, as I think they have done an outstanding job.

The article describes in detail the DmeRF system and its role in cobalt homeostasis in Vibrio parahaemolyticus. The article is aligned with the aims and scope of International Journal of Molecular Sciences.

In reference to the use of English, the style is appropriate and understandable. The number of tests done and results obtained is solid and adequate.

As for my proposed changes, I would remove all personal verb forms, such as "we used" in line 17, "we aimed" in line 73, "we demonstrated" in line 224, etc. The impersonal style would be more appropriate.

Another suggestion is to remove unnecessary capital letters in headings, including section subheadings. The titles should contain a brief overview of the content of the section, not its conclusion.

It is also noted that there are phrases in the introduction section, results and even figure captions that clearly belong to the discussion section. Examples of this are lines 73, 74, 146, 150, 162, 170, 180, 188, 191, 202. Much of the information on those lines is redundant with other information already contained in the discussion. I propose to eliminate the redundant information and relocate the one that corresponds to the discussion.

Round 2

Reviewer 1 Report

Initial suggestions (with new comments)

1. fig 4 could be supplemental data, and would be more meaningful with a diagram of the location of the primers and expected products in the wt and mutant loci. its not immediately clear what is expected and how the results support the expected locus structure.

A: Thank you very much for your advice. Correction has been made as suggested. Please refer to Figure S2 in the revised manuscript.

response: It's improved, but there is no scale on the diagram to judge product sizes (or indication of products and their size) so the utility is limited. In addition both wt and mutant loci should be shown (with expected products). If the data is included it should be useful and meaningful (an alternative is to leave out this figure entirely).

2. "complemented" DmeR strain (and other strains) are not actually complemented (i.e. phenotype restored to wild-type)

appears that "complemented strain is actually o/e strain - which is not ideal. need to add complemented strain with same expression as normal, or clarify that the "complemented" strain is an overexpression strain instead.

A: Generally, the gene deletion mutant harboring the pMMB207 plasmid containing the target gene is referred to as complementation strain. Please refer to [1-3] for examples. To our mind, an overexpression strain should be the WT strain harboring the pMMB207 plasmid expressing the target gene.

response - My suggestion in the original review was prompted by my initial confusion over the results of experiments using "complemented" strains. "complementation" implies (strongly, if not definitively) restoration of the approximate wt phenotype. In this case, the plasmid system used to "complement" the mutation increased expression substantially above what I would consider "WT", as evidenced by the data in figure 5 (revised). In fact the expression is over 500X higher in the complemented strain than in the wild-type! In this case, the presence or absence of the wild-type gene in strains carrying the plasmid has little effect on the final activity, and overexpression is the correct term to use. If the authors want to demonstrate "complementation", they need to use a plasmid that generates a near wild-type level of expression (within 2-fold is a good benchmark). Otherwise, they should interpret their results as a consequence of overexpression. This is not an academic concern - overexpression of another protein (e.g. another similar transporter), rather than the deleted gene, might generate a phenotype similar to their deletion; but this would be suppression, not complementation. Note that their observation that others have made the same mistake in the literature does not change my view.

3. On a related point, the supplemental expression data showing the effect of these manipulations (S2) should be in the main paper.

A: Thank you very much for your advice. Correction has been made as suggested. Please refer to Figure 5 in the revised manuscript.

4. The metal tolerance experiments shown in the supplemental data are problematic - they show clear effects of these genetic manipulations on zinc, cu and ni tolerance, but these effects are not really addressed by the authors. In particular the effect of the genetic changes (mutation/ overexpression) on Ni sensitivity clearly suggests a role for demRF in Ni homeostasis, but this is deemphasized by the authors. That data should be in the main paper, not in a supplement. In addition, some data for effect of these changes on Ni accumulation would be useful. ICP-MS analysis would reveal the full extent of these changes on elemental content and better support the authors arguments for specificity. As it is their data is compatible with a broad substrate specificity/regulatory response (fig 3) for this operon, but the authors seem focused on Co.

A: It is clear that the WT and mutant strains exhibited almost identical growth curves in the presence of other metals. However, the complementation strains exhibited decreased growth compared to the WT and mutant strains. In the complementation strains, the target gene is overexpressed; thus, the phenotype does not reflect the real function of the gene. The potential role of DmeF in Ni resistance has been discussed. Please refer to lines 281-284. 

 response - see my point above about overexpression. The authors are right to argue that their observations with the overexpressed strain are not physiological, i.e. not relevant to the normal function of the gene. This is why the effect of the plasmid must be attributed to overexpression, not complementation.

5. increased Co content does not prove that DmeF works via efflux - it could regulate influx by another factor for example. The authors should demonstrate their contention directly with efflux assays (loading cells with isotope to measure the rate of efflux, or by exchange with unlabeled Co).

A: We agree that increased Co content does not prove that DmeF works via efflux. In a previous study, the ΔtroRmutant of Streptococcus suis accumulated increased Mn and Co; we have never concluded that TroR works via efflux of Mn and Co [4]. Just as you said, TroR works via regulation of another factor. Nevertheless, in this study, DmeF has been annotated as CDF family Co(II)/Ni(II) efflux transporter; its homologues have been demonstrated to be Co (and Ni) efflux pump. Combined with these pieces of information and our findings, we concluded that TroR works via Co efflux. Similar claims have been found in other papers; please refer to [5-7] for examples.

response: The authors cannot discount the possibility that a transporter-like protein could evolve to become a sensor: in fact there are many examples of this (e.g. snf3 glucose sensor in yeast). The authors need to be more conservative with their conclusions here, unless they do the experiment I suggested. Again, others making the same mistake in the literature does not change my view.

6. In fig 7, authors use a different promoter fragment (gyrB) as a negative control, but the appropriate negative control is the demRF fragment with a scrambled demR binding site, to demonstrate specificity of binding. 

do the authors know what the binding site is? If not, they should perform a deletion analysis.

Likewise, showing mutation of the binding site eliminates demR function in vivo would be required support for their model (fig 9).

A: In the promoter of dmeRF, a DNA sequence flanked by two inverted repeats is predicted to be the binding site for DmeR. As suggested, the binding of DmeR to the demRF promoter without the binding site has been detected. Deletion of the binding site eliminates the binding capacity. Please refer to lines 176-177, 182-185 and Figure 8 in the revised manuscript.

I assume the above statement (binding of DmeR to the demRF promoter without the binding site has been detected ) is a typo? it seems to contradict the next statement.

- the figure is much improved, although ideally the site should have been mutated rather than deleted, to avoid changing spacing of other potential binding sites (if it was mutated/scrambled, please make that clear).

-fig 7 and 8 gels need to be a bit brighter to see the bands clearly.

V. parahaemolyticus DmeR shares low level of homology with DmeR from other species. The structural information of V. parahaemolyticus DmeR remains unknown. Therefore, we could not construct the mutant type DmeR that is non-functional.

response: An experiment showing that mutation/deletion of the dmeR binding site in DNA (not dmeR itself) eliminates DmeR function in vivo is required. 

7. If one of the main questions to address was what effect this operon has on infectivity, I don’t think the authors sufficiently tested this idea.

The authors don't report any role of this operon in infection, but the only output considered was death. perhaps a more detailed examination of the mice would reveal more subtle differences (for example, in bacterial load within infected animals). 

A: Actually, the mice is not a perfect model to evaluate V. parahaemolyticus virulence. This bacterium is difficult to colonize the tissues of mice. Death is usually the only output of infection. Please refer to [3,8,9] for examples. Due to COVID-19, we could not get qualified infant rabbits to evaluate the virulence. Therefore, we described that the DmeRF system is not required for V. parahaemolyticus virulence “in mice”. If conditions permit, further study will be performed to evaluate the virulence of these strains using the infant rabbit model.

8. It also seems as though the applied bacterial load was extreme, given the rapid death of the animals (all within a day) - perhaps an effect might have been seen if a lower inoculum were used, so that the mice were less severely affected.

A: Yes. All the mice died within 24 h. Generally, V. parahaemolyticus cause acute death of the animals in the mouse intraperitoneal infection model. Please refer to [3,8,9] for examples. A lower inoculum has been tested; however, only a few mice (less than 20%) died during the experiment.

response: points 7 and 8 - I think the above explanation suggests that it is possible to use the mouse model to more sensitively measure the effect of DmeRF mutation on virulence. i.e. the study should be designed so that not all the wt mice die quickly - this would potentially reveal smaller effects on virulence. So my criticism stands here. Nevertheless, since I have little experience with this vertebrate model, I leave it to the journal editor to decide if these results are sufficient to support the statement " not required for V. parahaemolyticus virulence “in mice”".

11. some supplemental figures are very low resolution and combined with the small data points are difficult to understand. better use of color and data point size is recommended.

A: In supplemental figure 3, the growth curves of the WT and mutant strains are almost identical; therefore, they are difficult to be distinguished. We have tried our best to improve the situation.

Unfortunately I can't see any change- needs a complete redo with larger/more distinctive data points and higher resolution.

Reviewer 2 Report

I am a bit disappointed that the authors have declined to pursue any of the metal quantitation experiments suggested in review and have discounted the possibility that their growth experiments may be showing interesting differences between strains, rather than "nearly identical". Even most of the minor suggestions to revise figures (resizing S3 for legibility and inverting the EMSA images) appear to have been ignored. The authors acknowledge that complemented strains are resistant to high nickel concentrations but have declined to follow up on this with Ni quantitation. The EMSA experiments with Cu and Zn were apparently performed and showed no binding to the promoter, but these are not mentioned in the revised manuscript.

Regarding growth curves, I understand that detailed statistical analyses of are not always performed. However, it is still possible to get a sense of variability. Rather than plotting a single experiment with error bars representing SD between three identical wells, why not plot the average at time points between experiments with a SDM? This may well show that the variability between experiments is too large to evaluate any differences in growth. Barring this, I think the authors would be better off to omit the data in figure S3 since it adds very little to the story as it stands.

All of that aside, it was my initial opinion that the manuscript has merit and that has not changed. I would like to see the data in figure S3 reported as described above (and resized for legibility) or omitted.

Author Response

We are grateful for your comments and suggestions. Measurement of intracellular Ni content in the strains have been performed during the first round of revision. However, the deadline was coming before we get the result. This result has been added to the revised manuscript, please refer to lines 158-159, 290-291, 364-372, and Fig. S3.

The adjustment for brightness and contrast has been tried for EMSA images, but they did not present well with a higher brightness and contrast.

Our study focuses on the role of DmeRF in Co homeostasis. The effect of Cu and Zn on dmeF expression may be due to an indirect influence of these metals on Co sensing. The EMSA experiments with Cu and Zn added little to our conclusion. Therefore, we did not mention it in the manuscript.

Yes, a number of papers plot data from three experiments for growth curves. Plotting a representative experiment is also alternative. Growth assays were performed for three to five times for the various conditions, not the same times. Therefore, we chose to plot a representative experiment; otherwise, the figure legends appear inconsistent when described the repetition.

For the first round of revision, we changed the y-axis of several figures in Figure S3, to make them larger. To further improve them, only the curves for the WT, ΔdmeF, and OEdmeF have been retained. Please refer to lines 133-139 and Figure S2 in the revised manuscript.

Round 3

Reviewer 1 Report

Thanks for your responsive changes.